

# The Cariboo Alpine Mesonet: Sub-hourly hydrometeorological observations of British Columbia's Cariboo Mountains and surrounding area since 2006

Marco A. Hernández-Henríquez[1], Aseem R. Sharma[2], Mark Taylor[1], Hadleigh D. Thompson[2], Stephen J. Déry[1]

[1]Environmental Science and Engineering Program, University of Northern British Columbia, 3333 University Way, Prince George, British Columbia, V2N 4Z9, Canada

[2]Natural Resources and Environmental Studies Program, University of Northern British Columbia, 3333 University Way, Prince George, British Columbia, V2N 4Z9, Canada

*Correspondence to*: Stephen J. Déry (sdery@unbc.ca)

**Abstract.** This article presents the development of a sub-hourly database of hydrometeorological conditions collected in British Columbia's Cariboo Mountains and surrounding area extending from 2006 to present. The Cariboo Alpine Mesonet (CAMnet) forms a network of 11 active hydrometeorological stations positioned at strategic locations across mid- to high elevations of the Cariboo Mountains. This mountain range spans 44,150 km$^2$ forming the northern extension of the Columbia Mountains. Deep fjord lakes along with old-growth redcedar and hemlock forests reside in the lower valleys, montane forests of Engelmann spruce, lodgepole pine and subalpine fir permeate the mid-elevations while alpine tundra, glaciers and several large icefields cover the higher elevations. The automatic weather stations typically measure air and soil temperature, relative humidity, atmospheric pressure, wind speed and direction, rainfall, and snow depth at 15 minute intervals. Additional measurements at some stations include shortwave and longwave radiation, near-surface air, skin, snow or water temperature, and soil moisture among others. Details on deployment sites, the instrumentation used and its precision, the collection and quality control process are provided. Instructions on how to access the database at Zenodo, an online public data repository, are also furnished (https://doi.org/10.5281/zenodo.1195043). Information on some of the

challenges and opportunities encountered in maintaining continuous and homogeneous time series of hydrometeorological variables and remote field sites is provided. The paper also summarizes ongoing plans to expand CAMnet to better monitor atmospheric conditions in BC's mountainous terrain, efforts to push data online in (near)real-time, availability of ancillary data, and lessons learned thus far in developing this

mesoscale network of hydrometeorological stations in the data-sparse Cariboo Mountains.

**Keywords:** hydrometeorological data, alpine meteorology, mountain hydrology, Cariboo Mountains, mesonet

## 1 Introduction

As in many regions worldwide, amplified climate change is altering the hydrometeorology of mountainous basins in northern British Columbia (BC). Climate change in north-central BC's mountains is accelerating permafrost degradation and thawing, snowpack thinning, glacier retreat, and decreasing the proportion of snowfall to total precipitation. The Cariboo Alpine Mesonet (CAMnet; MacLeod and Déry, 2007) is a mesoscale network (~100s km$^2$) initiated in 2006 by the Northern Hydrometeorology Group (NHG) at the

University of Northern British Columbia (UNBC) to collect long-term hydroclimatological data for the Cariboo Mountains and surrounding regions. Given the remote and difficult terrain in mountain environments where atmospheric and snowpack conditions vary substantially over small spatial scales, paucity of observational climate data exists in the world's alpine regions (Beniston, 2006; Bradley et al., 2004). This was indeed the case for the Cariboo Mountains where there is relatively good coverage of

meteorological stations below treeline elevation of 1700 m a.s.l.; however, coverage is particularly sparse in the higher elevations (Fig. 1). For example, there are currently only six active meteorological stations above treeline in the Cariboo Mountains, three of which are part of CAMnet. Thus, CAMnet fills an important observational gap to detect climate change at high elevations and its impacts to snowpacks, glaciers and water resources.

Since the summer of 2006, we have established over a dozen automatic weather stations and two radio repeater stations in valley and mountain settings, at elevations ranging from 683 to 2105 m a.s.l. (Déry et al., 2010). Data from these weather stations have supported modeling studies of seasonal snowpack evolution (Younas et al., 2017), blowing snow fluxes (Déry et al., 2010), turbulent fluxes on a mountain

glacier (Radić et al., 2017), glacial retreat (Beedle et al., 2009, 2015) and pro-glacial sediment transport dynamics (Leggat et al., 2015; Stott et al., 2016). These data have also been used to validate remote sensing products of snow (Tong et al., 2009a, b, 2010), gridded meteorological datasets (Sharma and Déry, 2016) and output from numerical weather prediction models over complex terrain (Schirmer and Jamieson, 2015). More recently, CAMnet stations have provided meteorological data to researchers investigating the

long-term physical, chemical and biological effects of the Mount Polley mine tailings impoundment breach in August 2014 (see Sect. 3.2) (Petticrew et al., 2015).

The purpose of this article is to document the development and current status of CAMnet including the setting in which the meteorological stations are deployed, the hydrometeorological variables being

monitored and collected, efforts to quality control the data, as well as data processing, archiving and availability. The paper also describes some of the challenges and opportunities incurred during the data collection process in the remote terrain of the Cariboo Mountains, the availability of ancillary data, and prospects for future expansion of the network in the Cariboo Mountains and beyond.

**2 Study area**

The Cariboo Mountains span ~44,150 km$^2$ in the central interior of BC, forming the northernmost range of the Columbia Mountains between the Interior Plateau and Rocky Mountain Trench (Fig. 2). Elevations in the Cariboo Mountains range from 330 m a.s.l. in the valley bottoms to 3520 m a.s.l. at its highest peak, Mount Sir Wilfrid Laurier (Sharma, 2014). The climate in the Cariboo Mountains remains drier than the

Coast Mountains to the west, but wetter than the Interior Plateau to the west and Rocky Mountains to the east (Beedle et al., 2015). On the windward side of the Cariboo Mountains in Quesnel, BC, the total annual precipitation averages ≈500 mm but increases to >2000 mm 150 km further east (Burford et al., 2009). Annual air temperature averages 1.8°C ± 0.96°C over the Cariboo Mountains region. Average annual total

precipitation in this region attains 739 mm ± 140 mm with nearly half falling as snow (Sharma, 2014). The relative amount of snow accumulation increases with altitude (Burford et al., 2009). Peak snow accumulation commonly surpasses 1000 mm water equivalent near treeline (Déry et al., 2014) and snowcover often persists from October through June (Tong et al., 2009a).

Within the North Cariboo region of BC lies the Quesnel River Basin, covering six biogeoclimatic zones over an area of ~12,000 km². The terrain varies from relatively flat to mountainous with a mean elevation of 1375 m a.s.l, rising from ~500 m a.s.l. in the west to ~3000 m a.s.l. in the northeast. Glaciers cover ~2% of the watershed while deep fjord lakes (e.g., Quesnel Lake with depths surpassing 500 m) reside in steep valleys. The mountainous terrain and snowy climate provide a suitable environment for the world's largest

Inland Temperate Rainforest (Déry et al., 2014; Farley, 1979). Situated along the Quesnel River sits UNBC's Quesnel River Research Centre (QRRC) that includes a large workshop, laboratory, office space, and housing residence. This unique combination of landscape and research infrastructure makes the watershed an ideal location for collecting high elevation hydrometeorological data.

## 3 Hydrometeorological stations

### 3.1 Overview

As of 31 March 2018, CAMnet comprises 11 active meteorological stations and two radio repeaters positioned at strategic locations, predominantly at mid- to high elevations within the Quesnel River Basin and the surrounding Cariboo Mountains of north-central BC (Figs. 3 and 4, Table 1). At each station, instruments measure atmospheric pressure, 2 m air temperature and relative humidity (with respect to

water), wind speed and direction, liquid precipitation, and snow depth among other variables that are only measured at specific stations (e.g., soil moisture, solar radiation, and near-surface air, snow, soil and water temperature) (Supplementary Table S1). Twelve-volt batteries charged during the day by solar panels power all sensors; however, both the QRRC and Ness Lake weather stations run on alternating current

(AC) supplemented by batteries in case of electrical interruptions. The QRRC station records incoming and outgoing shortwave and longwave radiation in addition to the parameters measured by the remote stations. Table 2 lists the specifications of each instrument, including their accuracy and precision. Dataloggers record the meteorological parameters and store data every 15 minutes. Stations located within the QRRC's proximity periodically send data via spread spectrum radio to the operational base computer located in the

housing residence (see Sect. 4.2).

**3.2 Chronological development**

Having identified a significant observational gap in the Cariboo Mountains and addressing the needs of the

QRRC's expanding research operations, UNBC's NHG established its first four CAMnet weather stations during the summer of 2006 in the vicinity of Likely, BC. Given the QRRC's availability to other UNBC researchers and those at other institutions, a dedicated effort was made to establish a standard meteorological station with an extended suite of instrumentation on the research centre's property. This station became operational on 11 August 2006 and is located 2 km southwest of Likely, BC in an incised

valley ~100 m south of the Quesnel River at an elevation of 743 m a.s.l. This is the only CAMnet weather station with a permanent 10-m tower with a concrete foundation for the tower footing and guy wires for anchoring. There is 50% sky visibility at the site and given its location, accessibility remains easy. Vegetation consists of grasses, redcedar, hemlock and some Douglas fir. A CR23X datalogger controls the entire weather station and accommodates the additional programming capacity and power supply needed to

run a full radiometer and associated heater, installed 5 m above bare ground. Winds are measured at 10 m

above the surface while air temperature and relative humidity are gauged 2 m above bare ground. This site also benefits from access to AC power to operate a heated precipitation gauge surrounded by an Alter shield. Data are periodically sent to the QRRC operational base computer via spread spectrum radio.

Three additional weather stations mounted on 3 m tripods were also deployed during summer 2006 near Likely, BC, in an attempt to capture the effects of elevation gradients on atmospheric conditions in the Cariboo Mountains. Factors leading to the selection of Spanish, Browntop and Blackbear Mountains as deployment sites included their range in elevations (1500-2000 m a.s.l.), accessibility (deactivated forestry roads and/or hiking trails), line-of-sight with each other, and one station with line-of-sight with the QRRC.

All were equipped with identical instrumentation to measure 2 m air temperature and relative humidity, 3 m wind speed and direction, atmospheric pressure, snow depth, rainfall, and soil temperature at 10-15 cm in depth. Stations at Spanish and Blackbear Mountains were installed in similar terrain (regenerating cutblocks of lodgepole pine stands) and elevations such that the meteorological conditions between the two exhibited little differences. As such, the meteorological instrumentation at Blackbear Mountain was

decommissioned during the summer of 2007 but the radio repeater was retained to act as the communication hub between the remote Spanish and Browntop Mountain stations and the operational base computer. The Browntop Mountain weather station lies at 2030 m a.s.l. in the alpine on an exposed ridge and endures excessive winds approaching 40 m s$^{-1}$ in some storms. These high winds have been quite challenging in keeping the entire station operational, with the continuity of readings frequently disrupted

through instrument damage (including the loss of multiple anemometers), icing and power malfunctions. A tipping bucket rain gauge was eventually dismantled on 11 July 2012 due to its low capacity to capture precipitation even when protected with an Alter shield. While a sonic ranger remains in operation at this site it rarely measures >10 cm of snow as winds continually erode it from the ridge top towards the lee side of Browntop Mountain where massive snowdrifts form and persist during summer.

Equipment previously installed at Blackbear Mountain was moved during August 2007 to a bedrock ridge at 2105 m a.s.l. near Castle Creek Glacier, 40 km west of McBride, BC, to support research of the Western Canadian Cryospheric Network. Despite its exposed and remote location, the station has experienced only a few minor issues mainly from strong winds affecting precipitation measurements, with the loss on several

occasions of the tipping bucket rain gauge's funnel and/or filter. A second weather station was installed at 1803 m a.s.l. in 2008, ~700 m in the Castle Creek Glacier forefield to complement the data being collected at higher elevation. The lower Castle Creek Glacier weather station is unique in the CAMnet network in measuring air temperature, relative humidity, wind speed and direction at two heights above the surface on a 6 m mast, allowing estimates of turbulent sensible and latent heat fluxes. This site also contains an array

of sensors to measure near-surface air or snow, skin surface and soil temperatures. A tipping bucket rain gauge protected by an Alter shield was added to the site on 21 August 2012. Both Castle Creek Glacier weather stations have line-of-sight with a radio repeater mounted in August 2009 on a bedrock ridge at 2219 m a.s.l. facing eastward towards Castle Creek Valley. Combined, the two weather stations at Castle Creek Glacier provide a nearly continuous record of atmospheric and snow conditions exceeding a decade

that complement well ongoing efforts to monitor the glacier's mass balance using push moraines, geodetic and remote sensing techniques.

A basic weather station was installed in 2007 for a period of two years at Mt. Tom, in the Cariboo Highlands near Wells and Barkerville, BC, to support the research projects of two UNBC graduate

students. This research was in collaboration with the BC Ministry of Forests, Lands, Natural Resource Operations and Rural Development's (FLNRORD's) Mt. Tom Adaptive Management Trial. The spatial distribution of snowcover in forested and clear-cut areas of different sizes as well as soil moisture memory were the focus of these two projects. Weekly snow survey data during one ablation season in addition to snow/surface temperatures were collected during this experiment.

Following the completion of the two graduate student projects at Mt. Tom, the equipment was transferred in October 2009 to a site at the Ancient Forest ~100 km east of Prince George at an elevation of 775 m a.s.l. The station was installed a few hundred meters from the Yellowhead Highway in a small clearing in the Inland Temperate Rainforest situated on the northwestern border of the Cariboo Mountains, now within

5    the boundary of the Ancient Forest/Chun T'oh Whudujut Provincial Park created in 2016. The damp, low-lying characteristics of this old-growth forest limit sky visibility and expose the site to abundant precipitation in all seasons with heavy snowfall during winter. Ground cover vegetation consists of large ferns, tall grasses, and dense Devil's club. Measurements at this site include soil moisture at three depths down to 65 cm. The CR10X data logger at this site was replaced with a CR1000 in August 2013 while the

10   tripod and a cross arm were replaced in August 2015 following a near direct hit by a felled old growth redcedar. This station provides critical information on BC's great 'snowforest' in a changing climate including support for ongoing ecosystem, ecological and outdoor recreation/tourism research at UNBC. It also serves routinely as a field site for a snow survey conducted by UNBC students enrolled in a course titled "Snow and Ice".

In proximity to the Ancient Forest, a basic meteorological station at Lunate Creek was acquired from the BC Ministry of FLNRORD in May 2010. The equipment at the station was fully updated and deployed in a regenerating redcedar/hemlock cutblock at an elevation of 953 m a.s.l. The tower sits on a 20° north-facing slope overlooking the Rocky Mountain Trench. Owing to bear and rodent activity along with the minimal

20   incoming solar radiation during winter this site has experienced periodic power interruptions and thus suffers extended data gaps. A second weather station was installed on a neighbouring cutblock at 1134 m a.s.l. to monitor upslope conditions during a period of one year in support of a graduate student project that quantified the contribution of snowmelt to soil moisture replenishment on the area's steep inclines. The toe slopes of the Cariboo Mountains are well known for their abundant soil moisture and springs even during

dry summer periods, with these water resources vital in sustaining the Ancient Forest's old growth forest stands.

Similarly to the Lunate Creek site, another weather station was acquired from BC Ministry of FLNRORD

in July 2010 at Lucille Mountain, west of McBride, BC. The weather station is deployed in a regenerating cutblock (planted) at an elevation of 1587 m a.s.l. and faces the Rocky Mountain Trench. Due to its difficult access and animal activity, the station has not been visited nor maintained since 2013.

Following the conclusion of the intensive field campaign at Lunate Creek, its upper station was transferred

in July 2012 to Ness Lake, ~35 km northwest of UNBC on the property of the senior/corresponding author. The weather station lies ~25 m from the eastern shore of Ness Lake at an elevation of 763 m a.s.l. The tower has only ~75% sky visibility since it is deployed in an area with large Douglas firs and two nearby residences. Data collected contribute to studying the effects that public recreational use has on Ness Lake wave activity and shoreline erosion. This site also serves to train students and research staff on use of the

meteorological equipment, testing new, damaged or repaired equipment, datalogger programming, and has also provided meteorological resources for UNBC courses. This site benefits from access to AC power and the Internet, facilitating automatic data transfers to a computer server at UNBC.

To support a graduate student's project investigating the water balance of a small boreal lake and its

watershed, a weather station was deployed at Coles Lake, ~100 km north of Fort Nelson, BC, near the border to the Northwest Territories in June 2013. The weather station resided ~500 m north of Coles Lake on the property of Quicksilver Inc., an oil and gas extraction company that, along with the BC Ministry of FLNRORD, supported the research. Apart from the meteorological data collected at the weather station, ancillary data included rainfall at three additional sites under various vegetation canopies, snow depth and

snow water equivalent through bi-weekly snow surveys at three sites during one ablation season, Coles

Lake inflows and outflows, shallow groundwater levels with three piezometer nests along the lakeshore, and lake skin surface water temperature.

Completion of the field campaign at Coles Lake allowed its station to be moved to the Kiskatinaw

Watershed in the summer of 2015. The site was located 300 m south of Arras Road and the John Hart Highway, ~19 km west of Dawson Creek, BC. The tower was installed in a ~4000 m$^2$ opening atop a small hill ~60 m north of the Hansen Reservoir. Ground cover consisted of unmaintained grass and the surrounding area is composed of mixed deciduous/coniferous trees and agricultural plots. Data for this station supported a graduate student's study on the prediction of the spring freshet in the Kiskatinaw River

that supplies domestic, commercial and industrial water for Dawson Creek and surrounding communities.

In a continuing effort to improve and integrate knowledge on the Nechako River (a major tributary to the Fraser River) and its watershed, a weather station was deployed at Tatuk Lake, BC, in September 2015. The weather station stands ~250 m from the northern shore of Tatuk Lake in the headwaters of the Chilako

River, ~75 km south of Vanderhoof, BC. It resides on Tatuk Lake Resort's private property, which provides safe and secure year-round access. The tower sits at an elevation of 938 m a.s.l. in an open section (~1000 m$^2$) where there is ~85% sky visibility. Tall grasses, shrubs, and rocky soil characterize the site. This station also serves as a reliable source of meteorological information for local communities of the Nechako Watershed and as a demonstration site for local elementary and middle school students.

The catastrophic failure of a tailings pond impoundment at Mount Polley Mine near Likely, BC, in August 2014 (Petticrew et al., 2015) renewed interest in monitoring closely atmospheric conditions at Quesnel Lake and surrounding area. While CAMnet included three weather stations near Likely, none was located on the shores of Quesnel Lake. To fill this observational gap, two lake-level sites were chosen for weather

station deployment in August 2016 and 2017. The Plato Point weather station is located 22 km southeast of

Likely on the sandy shoreline of Quesnel Lake just east of the Plato Island Resort. The tower sits at an elevation of 728 m a.s.l. and has ~90% sky visibility. The station has excellent exposure in all cardinal directions except towards the south, allowing measurement of long-fetch winds aligned along the principal axis of Quesnel Lake's main basin. During spring snowmelt, water levels rise partially submerging the

station. Two additional probes measure water and near-surface air temperatures. In the summer of 2017, the equipment from the Kiskatinaw Watershed weather station was moved to Long Creek, on a gravel beach along the shoreline of Quesnel Lake approximately halfway up its north arm. The tower is deployed at an elevation of 728 m a.s.l. and has ~75% sky visibility. Small shrubs and mature conifers flank the western side of the station that otherwise benefits from excellent exposure in all other cardinal directions.

Both the Plato Point and Long Creek weather stations allow detection of wind storms that generate basin-scale hydrodynamic processes in Quesnel Lake, leading to possible resuspension of mine tailings sediments deposited on the lake bottom during the 2014 spill.

### 3.3 Equipment used at each site

The majority of the meteorological equipment was purchased from Campbell Scientific Canada (CSC) and its suppliers; however, this report should not be construed as an endorsement of their products. Sole use of CSC equipment at nearly all CAMnet sites facilitates data cross comparisons, equipment substitutions, student and research staff training, and ensures the homogeneity of the hydrometeorological records (see Sect. 5.2). Most remote stations are assembled on 3 m tripods (i.e., CM110 or UT10) anchored to the

ground to maintain the orientation of the tower, ensure proper positioning of sensors, and prevent electrical damage from lightning strikes. Most of the sensors are installed and configured as recommended in the manuals (e.g., anemometers ~3 m above the ground surface and the air temperature and humidity sensors ~2 m above the ground surface). However, the precipitation gauges were an exception since the construction of concrete footings was not feasible at the remote sites. Instead, the precipitation gauges were

typically installed on 1.2 m × 1.2 m platforms anchored to the ground with rebar to support and level the

rain gauges as well as the Alter wind screens that reduce the effects of wind and turbulence on precipitation measurements. Most of the remote stations employ CR1000 dataloggers from CSC to power and operate the sensors. A CR23X datalogger is used at the QRRC station to run the sensors as it can provide the necessary power to engage the CNR1 net radiometer heater. Supplementary Table S1 provides a detailed

summary of the equipment and sensors used at each CAMnet station.

**3.4 Precision and accuracy of the instrumentation**

Table 2 provides the specifications, including precision and sensitivity, for sensors used at each CAMnet

station. These sensors are manufactured to withstand strong variations in environmental conditions and often remain unattended for lengthy periods of time. Most of the instruments, including the tipping bucket rain gauges, barometers and wind monitors, are initially calibrated and come with a certificate of accuracy. Depending on the amount of usage and the presence of adverse atmospheric conditions, some of the CAMnet instrumentation undergoes periodic recalibration, as maintaining the precision and quality of data

remains of utmost importance to the development of the database. Recalibration and regular maintenance are completed if the quality of the data being sampled starts to become affected or if there is obvious damage to the instrumentation. Common errors in sampling precision and accuracy stem from worn out equipment such as the depth-to-target data recorded by the SR50 ultrasonic depth sensor that can contain outlying values when the transducer wears out. This causes peaks in the depth-to-target data that are carried

over to the snow depth data through calculations. These peaks can be filtered out relatively easily so they do not significantly affect data quality.

All instrumentation used at CAMnet weather stations include a recommended temperature operating range where data collected outside these parameters is subject to larger errors. However, these operating ranges

are quite vast (typically -40°C to +70°C) thus minimizing potential errors. Of note, dataloggers with

extended temperature operating ranges are employed at all CAMnet sites given the frigid conditions often

encountered during winter in the Cariboo Mountains and surrounding area. Snow and ice, like temperature,

have the ability to affect the precision of the data and accuracy of the equipment. Rime build up can cause

the anemometer blades to freeze up resulting in extended periods of data with wind speeds of 0 m s$^{-1}$. Snow

accumulation on pyranometers and the radiation shields of temperature/relative humidity probes can also

affect the precision of the sampled data. Regular site visits mitigate these issues and allow for regular

instrument check-ups to maintain the quality of the data.

## 4 Data collection

### 4.1 Frequency

Dataloggers were programmed using the Loggernet SCWin, CRBasic, and Edlog software from CSC. All

of the station programs employ scan intervals of one minute and data intervals of 15 minutes. Depending

on the sensors and the specific variable being measured, sampling either occurs once either every 15

minutes or is measured for the entire data interval and averaged. Some of the measurements are also simply

summed over the 15 minute period, such as precipitation and incoming solar radiation. Regardless of the

sampling interval, all collected data are stored in 15 minute timestamps in the datalogger memory and the

database spreadsheets.

### 4.2 Automated data transfers

Spread-spectrum radios facilitate communication and automated data transfers between the weather

stations at Spanish Mountain, Browntop Mountain and Blackbear Mountain (no longer equipped with

meteorological equipment) with the operational base computer. Spread-spectrum radios have a range of

~30 km within line-of-sight; however, obstructions such as hills and vegetation attenuate significantly the

radio signals while the specific type of antenna used modulates signal strength. Thus a radio signal range of

~13 km was chosen during the site selection process for stations 1-4 (Table 1). The Blackbear Mountain

radio repeater routes radio communication from the Spanish Mountain and Browntop Mountain stations

once daily to the QRRC operational base computer that acts as a gateway to online data retrieval. Radio
communication from the QRRC weather station to this computer occurs on an hourly basis allowing online
data transfers to a computer server on the UNBC campus in Prince George, BC, for web-based
visualization and long-term archiving.

A second radio repeater station installed on a bedrock ridge at Castle Creek Glacier with line-of-sight with
the two nearby weather stations and the Castle Creek Valley allowed remote data downloads at the end of a
deactivated forestry road accessed by a 4×4 vehicle. A major landslide in 2012, however, has since blocked
road access to the upper reaches of the Castle Creek Valley rendering this radio repeater obsolete. Thus the
radio repeater station at Castle Creek Glacier may soon be relocated to a site near Quesnel Lake to extend
remote communication at sites there.

### 4.3 Gaps and infilling

Some time series within the database have gaps (Fig. 5), either for entire stations or individual instruments,
and for a variety of reasons. Stations based in wilderness settings have experienced interruptions due to
environmental factors such as falling trees, intense storms, and wintertime icing (see ECCC, 2015). Less
common, but potentially more destructive to the equipment and instrumentation, involve wildlife
interactions, particularly with large mammals such as black and grizzly bears. Since the NHG's research
requires long-term historical climate records, it is necessary to find other resources that may fill some of
the data gaps. A few meteorological stations exist in the vicinity of the Quesnel River Basin that may be
able to provide additional data to complete this record (Fig. 2). Specifically, the BC Ministry of
FLNRORD Likely Aerodrome weather station (52°36'05" N, 121°30'48" W, elevation 1046 m a.s.l.), the
BC Ministry of Environment snow pillows at Yanks Peak East (52°49' N, 121°21' W, elevation 1683 m
a.s.l.) and Barkerville (53°03' N, 121°29' W, elevation 1520 m a.s.l.), and the Environment and Climate
Change Canada (ECCC) Barkerville weather station (53°04'09" N, 121°30'53" W, elevation 1283 m a.s.l.)

have reliable and relatively long-term meteorological measurements for the Cariboo Mountains and in the vicinity of CAMnet weather stations. Of note, the ECCC Barkerville record begins in 1888, making it BC's third longest observation-based meteorological dataset; however, this remains a weather station operated by volunteers rendering data quality and homogeneity suspect at times. Nonetheless, these proximal

weather and snow pillow stations provide valuable data on hydrometeorological conditions when CAMnet stations experienced data gaps. CAMnet stations are notably deficient in measuring cold season precipitation and snow water equivalence, variables that may be acquired from these additional weather stations.

Temporal gaps at one CAMnet station may also be infilled with data from another proximal one in the network. For instance, daily air temperatures for the QRRC, Spanish Mountain, and Browntop Mountain weather stations are highly correlated ($r \geq 0.92$; Sharma, 2014). The local surface air temperature lapse rates vary from month to month with the steepest rates (6.6°C km$^{-1}$) during summer and with more modest levels during winter (2.7°C km$^{-1}$) (Sharma, 2014). Indeed, potent air temperature inversions often arise in

winter such that care must be exercised in reconstructing this variable. Vigorous orographic enhancement of precipitation occurs in the Cariboo Mountains inducing strong precipitation gradients across their steep elevation gradients. Thus site-specific conditions such as altitude, exposure, land and vegetation cover induce considerable spatial variations in all quantities measured at CAMnet weather stations that require special attention in reconstructing missing data.

### 4.4 Data quality assessment and control

For CAMnet observed data, quality assessment is carried out for individual parameters at each station through statistical quality control procedures, analysis of frequency intervals, values, manual observations, etc. The CAMnet database consists of the original raw data allowing users to manipulate it according to

their needs and desires. Missing timestamps are identified and filled with "NA" using codes developed in R

(R Core Team, 2014) to maintain a complete yearly data record for each station. All data values assessed as erroneous, extreme, or outside a predetermined frequency interval are simply flagged (highlighted in yellow) and recorded in the metadata documentation. Data from several sampled variables are also assessed for their quality through visual inspection and scatter plots such as snow depth and soil

temperature. An outline of this quality assessment and control techniques can be found in a document located in the online database in each of the station's respective folders.

## 4.5 Metadata

Metadata for each climate station in the CAMnet network are important to take into consideration when

utilizing the associated data and are therefore summarized in a detailed .docx file. This document, a copy of which is available with the data for each station, outlines site location (coordinates, elevation, etc.) and access, provides a brief overview of the landscape, station equipment specifics (e.g., instrument heights) and data collection parameters.

Field notes and site logs are documented using a mediawiki.org webpage allowing the ability to check on historical station visits dating back to the original deployment. These logs remain particularly valuable due to the high turnover of students and research staff in the NHG (e.g., summer field assistants) and serve as a complete and detailed resource. Weather conditions and equipment status, as well as work carried out, comprise important factors noted during each station visit and are therefore documented in these logs.

Moreover, instrument serial numbers, recalibration and servicing schedules as well as an ongoing 'to-do' list of field work activities are systematically updated on the CAMnet wiki.

## 5 Challenges and opportunities

### 5.1 Physical and environmental challenges

The construction and maintenance of a highly sensitive, long-term database such as CAMnet generates
distinct challenges on a variety of scales. Power outages due to battery failure/solar panel issues are a
major contributor towards data loss and interruptions in sampling (see Sect. 4.3), a risk which is higher
during the winter season when snow and ice can cover solar panels and immerse instrumentation while

cold temperatures may accelerate the depletion of batteries. CAMnet stations have also been plagued with
several animal incidents whereby the equipment has been displaced, damaged, or rendered inoperable.
Additionally, wildlife also has the ability to influence data solely by its presence, with snowpack
measurements that rely on an undisturbed section of snow beneath the sensor, being especially vulnerable.
Other environmental considerations include lightning strikes that are mitigated by robustly grounding each

station, and the risk of damage from wildfires, which were extensive throughout the Cariboo region during
the summer of 2017 and led to ashfall in nearby precipitation gauges. Physical challenges are presented by
the complex terrain and remote locations of many stations, which also preclude frequent site visits; some
stations such as those situated near Castle Creek Glacier are only visited once per year. Methods to access
station sites include 4×4 vehicle (often on deactivated logging roads), helicopter, on foot (hiking in

summer, snowshoeing in winter), and snowmobile. Since these remote locations and environments are
fundamental for local and regional hydrology, the importance for hydrometeorological data collection and
the CAMnet network is pronounced; however, difficulty associated with securing funds to sustain this
extensive network of remote weather stations (e.g., costs associated with field work and travel, equipment
repairs, recalibration and purchases, student and research staff wages, etc.) remains a distinct and ongoing

challenge.

### 5.2 Data homogeneity challenges

To minimize possible spurious trends or step changes in the hydrometeorological observations, all CAMnet
sites are inspected and maintained regularly with faulty instrumentation repaired or replaced as necessary.

While dedicated efforts are made in substituting deficient equipment with the same make and model, this is

not always feasible if, for instance, the supplier no longer fabricates a given piece of equipment or specific

model. Meteorological towers may shift under the influence of winds, frost heaving and heavy snowpacks,

altering the measurement heights and possibly rendering the station unleveled. Modifications to datalogger

programs (e.g., to switch on or off the heater for the QRRC radiometer) or operating systems may induce

additional discontinuities in the CAMnet data.

Despite efforts to maintain consistency in the CAMnet datasets by routine station inspection and

maintenance, various factors have resulted in a small fraction of data being influenced by external factors.

Examples of these include the encroaching vegetation around the station on Spanish Mountain, the

accidental rotation of the Browntop wind vane (away from north), and the replacement of instruments and

equipment after various wildlife incidents and severe storms. As with data gaps, no post-processing of the

data has been performed to assess their homogeneity, to remove suspect trends or erroneous data points.

Site photos, field notes, and post-trip updates of metadata remain invaluable resources in resolving issues

related to these external influences, and are available for any parties utilizing CAMnet data.

### 5.3 Opportunistic measurements

A primary objective of CAMnet remains to develop long-term hydrometeorological records in the poorly

sampled Cariboo Mountains to assess climate change impacts on regional snow, glacier and water

resources. This requires long-term homogeneous records of hydrometeorological conditions in the Cariboo

Mountains at strategic locations such as the QRRC, Browntop and Spanish Mountains, and Castle Creek

Glacier. A broader role of this long-term monitoring is the ability to answer unforeseen questions or

research needs such as wildfires, pest infestations, mine spills (such as the Mount Polley Mine accident),

future resource extraction, and land use/cover change in the area in addition to climate change. Indeed,

CAMnet may assist answering questions that have not yet been asked or even contemplated while

providing baseline data in the event of future disturbances. Pushing CAMnet data online in (near)real-time

also allows monitoring current atmospheric conditions remotely to assist mitigating extreme events such as atmospheric rivers, floods or intense convective storms that can ignite wildfires such as in July 2017 in the BC Interior. Finally, the mobile nature of the 3 m masts facilitates the transport and rapid deployment of weather stations to various sites, often to support a graduate student's research project or an intensive field

campaign at a designated site.

### 5.4 Research, training and educational opportunities

CAMnet now assembles over a decade of sub-hourly hydrometeorological observations in the Cariboo Mountains and surrounding area, forming the core data for multiple undergraduate and graduate student

projects at UNBC and other institutions. It allows training opportunities for students and research staff to develop skills in deploying meteorological instruments, datalogger programming, data quality control and interpretation, as well as field expertise in remote terrain. CAMnet data and site photos are routinely used in undergraduate and graduate level courses to demonstrate regional examples of phenomena studied in class. CAMnet weather stations also provide staging sites for educational videos (e.g., for the Vancouver

Aquarium's Year of Science series), for media interviews and for outreach activities for students of all levels. Finally, it provides opportunities to undertake interdisciplinary research at UNBC and institutions across Canada and abroad.

### 6 Ancillary data

Other studies focused on the Cariboo Mountains and surrounding area have generated ample ancillary data

to the CAMnet hydrometeorological records. This includes an ongoing effort to collect seasonal/annual mass balance measurements at Castle Creek Glacier initiated in 2008 by colleagues at UNBC. This site is of particular interest given its record of annual push moraines allowing a reconstruction of the glacier's retreat since the mid-1940s (Beedle et al., 2009). Eddy covariance measurements of turbulent fluxes on

Castle Creek Glacier were also conducted during two summer field campaigns (in 2010 and 2012) during which additional atmospheric measurements were collected on the glacier (Radić et al., 2017). Sediment fluxes in Castle Creek and its main tributaries were also sampled during two other summer field campaigns (in 2008 and 2011; Leggat et al., 2015; Stott et al., 2016). Meteorological observations along the shores of

Quesnel Lake support ongoing efforts to understand the long-term impacts of the Mount Polley Mine tailings pond spill on water quality and ecology of the system (Petticrew et al., 2015). A dedicated effort, based at the QRRC, is being made in monitoring lake conditions (e.g., vertical profiles of water temperature, electrical conductivity, turbidity, etc.), as well as downstream water quality, temperature, and sediment concentrations and quality in the Quesnel River. This ongoing project benefits from historical,

background data collected on Quesnel Lake starting in the early 2000s (Laval et al., 2008). Finally, provincial and federal networks operate additional meteorological and snow pillow stations while the Water Survey of Canada manages several hydrometric gauging stations on the main waterways draining the Cariboo Mountains including the Fraser, North Thompson, Clearwater, Quesnel, Horsefly, Doré, Willow and Bowron Rivers (see Fig. 2 and Déry et al., 2012). This assemblage of unique ecosystems in

pristine landscapes, availability of existing hydrometeorological data and a reliable research station to base field activities make the Cariboo Mountains particularly attractive for ongoing environmental and ecological research.

**7 Future expansion and application of CAMnet**

During the summer of 2018, two to three additional meteorological stations will be deployed along the shores of Quesnel Lake in support of an ongoing project investigating its resiliency to the Mount Polley mine tailings impoundment breach. Quesnel Lake has a complex morphometry with the steep surrounding topography often channeling winds. The area is also susceptible to mountain-valley circulations during clear summer days, with katabatic winds descending the steep, glacierized and snow-covered slopes

towards the lower reaches of the valleys. Concerns over resuspension of lake-bottom sediments deposited

during the tailings pond breach associated with wind-driven seiches requires improved monitoring of atmospheric conditions including winds along Quesnel Lake.

In support of a new project titled the "Storms and Precipitation Across the continental Divide Experiment"
(SPADE), a CAMnet weather station will be deployed near or within Kootenay National Park in southeastern BC during the fall of 2018. This will complement an existing array of meteorological stations operated by the University of Saskatchewan's Coldwater Laboratory at Fortress Mountain and Marmot Creek, Alberta, on the eastern flanks of the Canadian Rockies (Rothwell et al., 2016) and by the University of Calgary at Haig Glacier on the North American continental divide (Shea et al., 2005). An intensive field
campaign will be conducted during May and June 2019 to obtain high temporal resolution atmospheric conditions with an emphasis on precipitation on a longitudinal transect across the continental divide.

**8 Data availability**

Data described in this article may be downloaded from Zenodo (https://doi.org/10.5281/zenodo.1195043).
CAMnet meteorological data are available as Excel spreadsheets in .xlsx format. The database comprises
of a specific directory for each station that includes yearly .xlsx data files named according to the site location and year, and a .docx file outlining metadata describing the location, quality control performed to evaluate the data, installed instruments and sampling parameters of each instrument per station. Data files are listed chronologically and follow the numerical order listed in Table 1.

**9 Conclusions**

CAMnet was initiated from an interest and need to collect high temporal resolution data in elevated, complex terrain, in a mountainous region where hydrometeorological data were previously quite sparse. Although challenges abound obtaining high quality data in a wilderness setting, our ever-expanding hydrometeorological datasets provide insight to the weather and climate of the Cariboo Mountains and

surrounding region. We have presented an overview of CAMnet station locations, the instruments used, parameters measured, and the resulting datasets, since 2006. The planned addition of stations around Quesnel Lake and along the continental divide demonstrate the ongoing expansion of the network, while an extensive collection of metadata contributes to the understanding of data inconsistencies and possible

errors. We encourage interested parties to contact the NHG for information on using CAMnet data, or further discussion on the establishment and maintenance of a mesoscale network of hydrometeorological stations.

**Author contributions.** SJD initiated the development of CAMnet and designed the present report. MAHH

wrote the first draft of the manuscript with contributions from ARS, MT, HDT and SJD, and all contributed to the discussion of the database and manuscript refinement.

**Competing interests.** The authors declare they have no conflicts of interest.

**Special issue statement.** This article forms a contribution to the special issue "Hydrometeorological data from mountain and alpine research catchments" edited by Dr. John Pomeroy and Dr. Danny Marks. It is not associated with a conference.

**Acknowledgements.** Thanks to UNBC, Future Forest Ecosystems Scientific Council, Environment and

Climate Change Canada, Natural Sciences and Engineering Research Council of Canada through the Canadian Sea Ice and Snow Evolution network and the Discovery Grant and Research Tools and Instrumentation programs, Nechako Environmental Enhancement Fund, Canada Foundation for Innovation, Canadian Foundation for Climate and Atmospheric Sciences through the Western Canadian Cryospheric Network, BC Knowledge Development Fund, Global Water Futures and the Canada Research

Chair program of the Government of Canada for funding. Thanks to Sam Albers, Michael Allchin, Darwyn Coxson, Art Fredeen, Scott Green, Rick Holmes, Peter Jackson, Brian Menounos, Ken Otter, Phil Owens, Margot Parkes, Ellen Petticrew, and Roger Wheate (all previously or currently affiliated with UNBC),

Bernard Laval (UBC), Vanessa Foord, Richard Kabzems and Pat Teti (FLNRORD), and many field assistants and volunteers for their contributions. Thank you to the staff at the QRRC (including past managers Rick Holmes and Sam Albers, and current manager Michael Allchin) and the people of Likely and area for their continued support and interest in our research. Constructive and insightful comments

5   from Phil Owens (UNBC) led to a much improved paper.

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



**Table 1.** Details for each CAMnet station.

| Station | Name | Latitude (°N) | Longitude (°W) | Elevation (m a.s.l.) | Terrain | Date of Deployment | Date of Decommission |
|---|---|---|---|---|---|---|---|
| 1 | QRRC[a] | 52°37'06" | 121°35'24" | 743 | Flat short grass | 11 Aug. 2006 | |
| 2 | Spanish Mountain | 52°33'48" | 121°24'35" | 1509 | Sloped forest regrowth | 30 June 2006 | |
| 3 | Browntop Mountain | 52°42'28" | 121°20'02" | 2030 | Alpine ridge | 23 Aug. 2006 | |
| 4 | Blackbear Mountain Radio Repeater[b] | 52°36'53" | 121°26'12" | 1590 | Sloped forest regrowth | 24 Aug. 2006 | |
| 5 | Upper Castle Creek Glacier | 53°02'36" | 120°26'18" | 2105 | Bedrock ridge | 28 Aug. 2007 | |
| 6 | Mount Tom | 53°11'32" | 121°39'49" | 1490 | Sloped cutblock | 20 Sept. 2007 | 15 Sept. 2009 |
| 7 | Lower Castle Creek Glacier | 53°03'45" | 120°26'04" | 1803 | Flat moraine | 13 Aug. 2008 | |
| 8 | Castle Creek Glacier Radio Repeater | 53°02'03" | 120°25'57" | 2219 | Bedrock ridge | 1 Aug. 2009 | |
| 9 | Ancient Forest | 53°46'21" | 121°13'44" | 774 | Old-growth forest | 7 Oct. 2009 | |
| 10 | Lunate Creek | 53°49'56" | 121°27'32" | 953 | Regenerating clearcut | 17 May 2010 | |
| 11 | Lucille Mountain | 53°16'22" | 120°14'18" | 1587 | Planted cutblock | 7 July 2010 | 10 July 2013 |
| 12 | Upper Lunate Creek | 53°50'02" | 121°28'21" | 1134 | Regenerating clearcut | 23 Aug. 2011 | 29 July 2012 |
| 13 | Ness Lake | 54°01'28" | 123°05'36" | 763 | Flat short grass | 30 July 2012 | |
| 14 | Coles Lake | 59°47'22" | 122°36'43" | 479 | Flat - bulldozed | 11 June 2013 | 29 July 2015 |
| 15 | Kiskatinaw | 55°45'04" | 120°31'27" | 726 | Flat long grass | 20 Aug. 2015 | 15 June 2017 |
| 16 | Tatuk Lake | 53°32'41" | 124°16'36" | 938 | Flat short grass | 29 Sept. 2015 | |
| 17 | Plato Point | 52°29'15" | 121°17'03" | 728 | Sandy beach | 17 Aug. 2016 | |
| 18 | Long Creek | 52°40'07" | 120°57'34" | 728 | Gravel beach | 26 June 2017 | |

[a] Quesnel River Research Centre. [b] The Blackbear Mountain weather station was dismantled in July 2007 and became a radio repeater station to facilitate communication and data download at the QRRC.



**Table 2.** Specifications for sensors used at each CAMnet station.

| Sensor | Model | Measurements | Units | Sensitivity | Accuracy | Operating Range |
|---|---|---|---|---|---|---|
| **Spread Spectrum Radio** | RF401 | N/A | N/A | –109 dBm | N/A | –25°C to 50°C |
| **Barometric Pressure Sensor** | 61205V 61205V-10 | Atmospheric Pressure | hPa | 0.1 hPa | ± 0.5 hPa | 500-1100 hPa |
| | 61302V | Atmospheric Pressure | hPa | 0.1 hPa | ±0.2 hPa (25°C) ±0.3 hPa (-50 to +60°C) | 500-1100 hPa |
| | CS106 | Atmospheric Pressure | hPa | N/A | ±0.3 hPa (20°C) ±0.6 hPa (0°C to 40°C) ±1.0 hPa (-20°C to +45°C) ±1.5 hPa (-40°C to +60°C) | –40°C to 60°C  500-1100 hPa |
| **Relative Humidity & Air Temperature Probe** | HMP35C HMP45C HMP45C212 | Air Temperature | °C | N/A | ± 0.1°C | –40°C to 60°C |
| | | Relative Humidity | % | N/A | ± 2% (0-90% RH) at 20°C ± 3% (90-100% RH) at 20°C | |
| | HMP60 | Air Temperature | °C | N/A | ±0.6°C (-40° to +60°C) | |
| | | Relative Humidity | % | N/A | ±3% RH (0-90% RH) ±5% RH (90-100% RH) | |
| | HC-S3 | Air Temperature | °C | N/A | ±0.2°C (-30°C to 60°C) | –40°C to 60°C |
| | | Relative Humidity | % | N/A | ±1.5% at 23°C | |
| **Temperature Probe** | 107B | Soil Temperature | °C | N/A | ±0.4°C (–24°C to 48°C) ±0.9 °C (–35°C to 50°C) | –50°C to 100°C |
| | 109 109B | Soil Temperature | °C | N/A | ±0.60°C (–50°C to 70°C) ±0.25°C (–10°C to 70°C) | |
| **RM Young Wind Monitor** | 05103-10 05103-45 | Wind Speed | m s⁻¹ | N/A | ± 0.3 m s⁻¹ | –50°C to 50°C |
| | | Wind Direction | degrees | N/A | ±3° | |
| **Tipping Bucket Rain Gauge** | 34-HT-P | Liquid/Solid Precipitation | mm | 0.25 mm per tip | Precipitation Rate:  ±0.25 mm up to 20 mm hr⁻¹ ±3% over 20 mm hr⁻¹ | –20°C to 50°C |
| | TE525WS | Liquid Precipitation | mm | 0.254 mm per tip | Precipitation Rate:  ±1% up to 25.4 mm hr⁻¹ +0, –2.5 % from 25.4 to 50.8 mm hr⁻¹ +0, –3.5 % from 50.8 to 76.2 mm hr⁻¹ | 0°C to 50°C |
| | TE525M | Liquid Precipitation | mm | 0.1 mm per tip | Precipitation Rate:  ±1 % up to 25.4 mm hr⁻¹ +0, –2.5 % from 25.4 to 50.8 mm hr⁻¹ +0, –3.5 % from 50.8 to 76.2 mm hr⁻¹ | |
| | TR-525USW | Liquid Precipitation | mm | 0.01 in per tip | Precipitation Rate:  1% at 25.4 mm hr⁻¹ or less | |



| | | | | | | |
|---|---|---|---|---|---|---|
| **Ultrasonic Distance Measurement** | SR50 | Snow Depth | cm | 0.1 mm | ±1 cm or 0.4% of distance to target (whichever is greater) | -30°C to 50°C |
| | SR50A | Snow Depth | cm | 0.25 mm | ±1 cm or 0.4% of distance to target (whichever is greater) | -45°C to 50°C |
| **Water Content Reflectometer** | CS616 | Soil Moisture | % VWC | <0.1% VWC | ±2.5% VWC using standard calibration with bulk electrical conductivity ≤0.5 dS m$^{-1}$ and bulk density ≤1.55 g cm$^{-3}$ in measurement range 0% to 50% VWC | 0°C to 70°C |
| | CS650 | Soil Moisture | % VWC | <0.05% VWC | ±3% (typical with factory VWC model) where solution EC < 3 dS/m | -50°C to 70°C |
| **Radiometer/ Pyranometer** | CNR1 | Shortwave Up/Down | W m$^{-2}$ | 6.18 µV/W/m$^2$ | ± 10% of daily totals | -40°C to 70°C |
| | | Longwave Up/Down | | | | |
| | CMP3 | Shortwave Up/Down | W m$^{-2}$ | 5 to 20 µV/W/m$^2$ | ± 10% of daily totals | -40°C to 80°C |
| | | Longwave Up/Down | | | | |
| | SP LITE2 | Shortwave Up/Down | W m$^{-2}$ | 81 µV/W/m$^2$ <±2% shift per year | ± 5% of daily totals | -40°C to 80°C |
| | | Longwave Up/Down | | | | |
| | LI 200R | Shortwave Up/Down | W m$^{-2}$ | 0.13 kW m$^{-2}$ mV$^{-1}$ | Absolute error in natural daylight is ±5% maximum; ±3% typical | -40°C to 65°C |
| | LI 190SB | Longwave Up/Down | | 5 µA per 1000 µmoles s$^{-1}$m$^{-2}$ | Maximum deviation of 1% up to 10,000 µmoles s$^{-1}$m$^{-2}$ | |
| **Precision Infrared Radiometer** | SI-111 (IRR-P) | Surface Temperature | °C | ±0.1°C @ −10° to 65°C ±0.3°C @ −40° to 70°C | ±0.2°C @ −10° to 65°C (where target temperature is within 20°C of sensor body temperature) ±0.5°C @ −40° to 70°C (where target temperature is >20°C of sensor body temperature) | −55° to 80°C 0 to 100% RH |




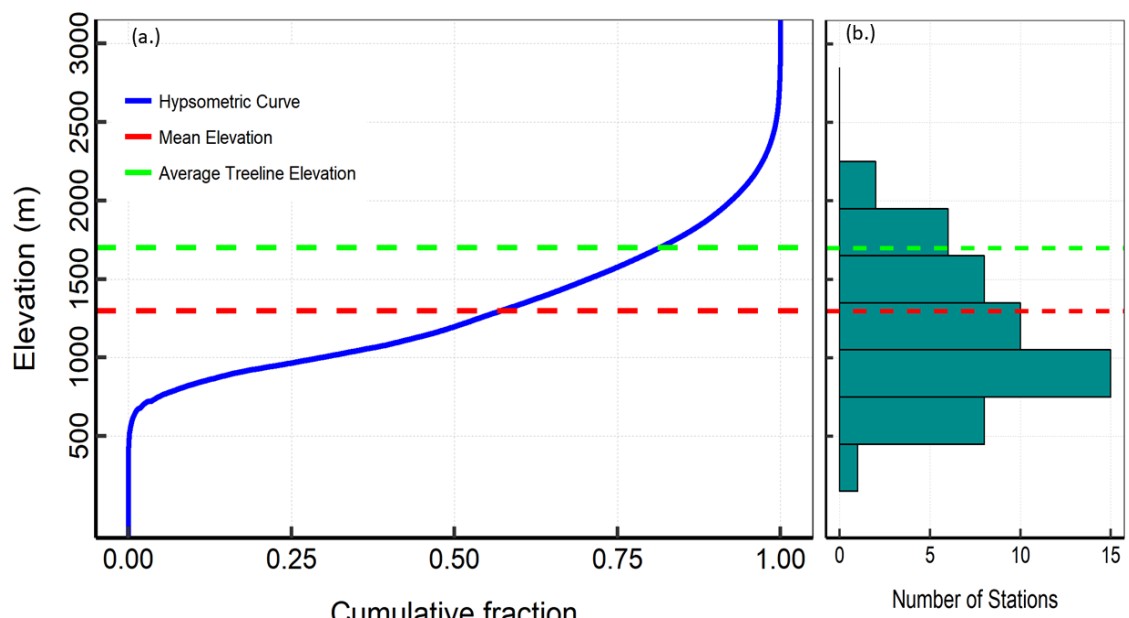

**Figure 1.** (a) Hypsometric curve showing the cumulative fraction of the elevation ranges in the Cariboo Mountains Region. (b) The number of active weather stations at different elevations within the Cariboo Mountains Region.



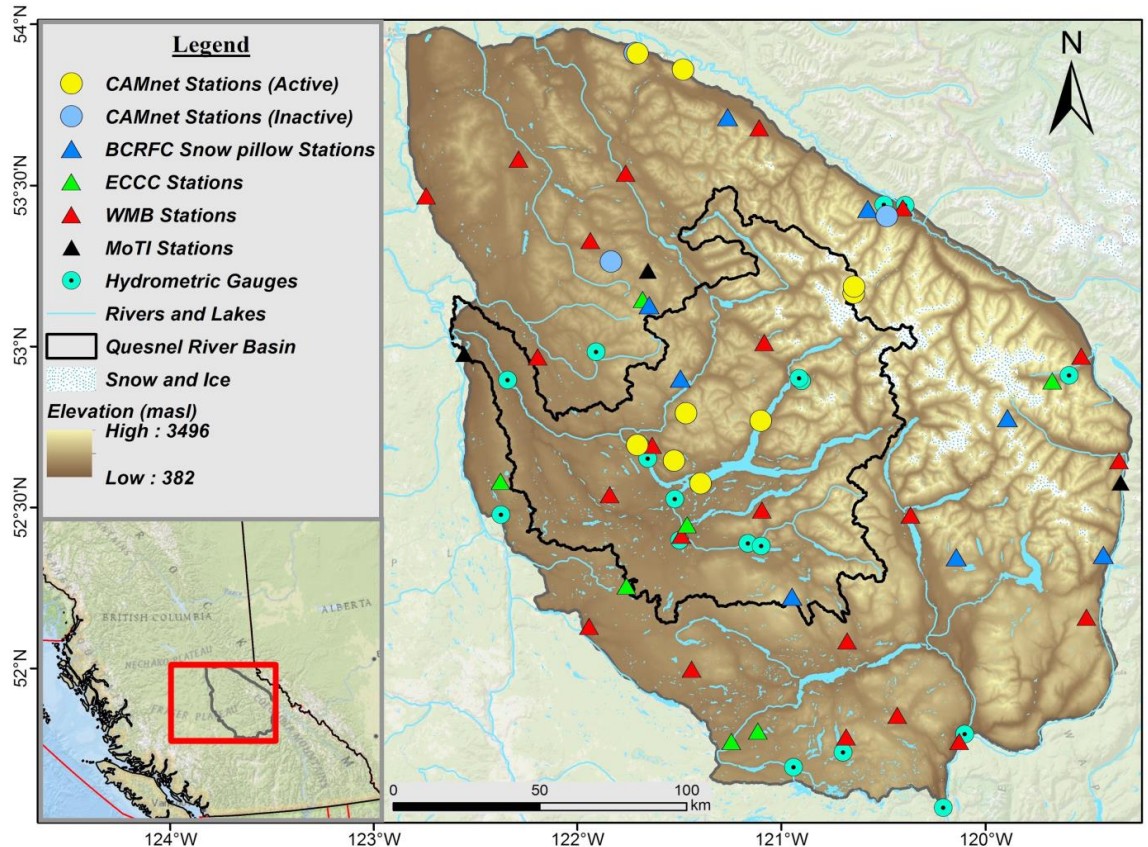

**Figure 2.** Topographic map of the Cariboo Mountain Region including the Quesnel River Basin. Dots and triangles show the location of active weather, snow pillow and hydrometric stations operated by different agencies in this region including the BC River Forecast Centre (BCRFC), Environment and Climate Change Canada (ECCC), BC Wildfire Management Branch (WMB), and BC Ministry of Transportation and Infrastructure (MoTI).





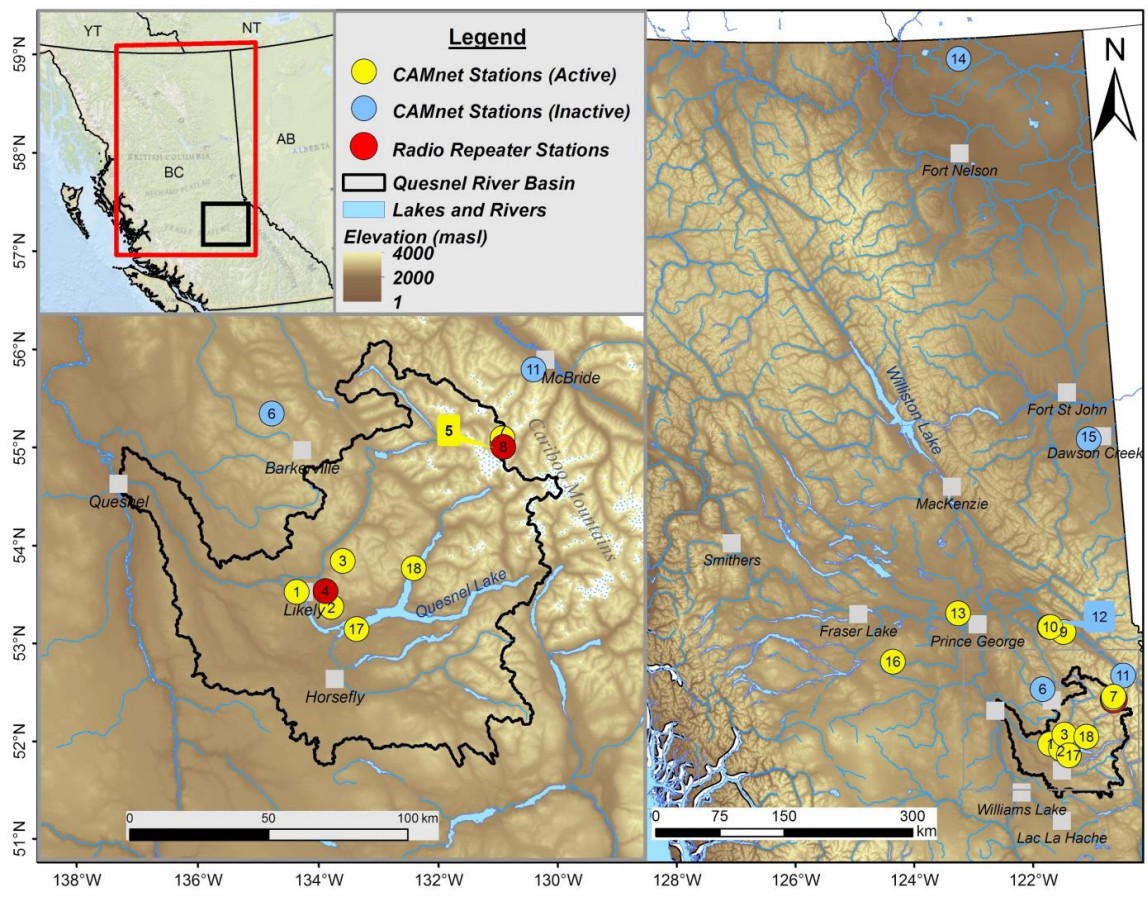

**Figure 3.** Map of north-central BC showing the location of active and inactive CAMnet weather and radio repeater stations, numbered as in Table 1.





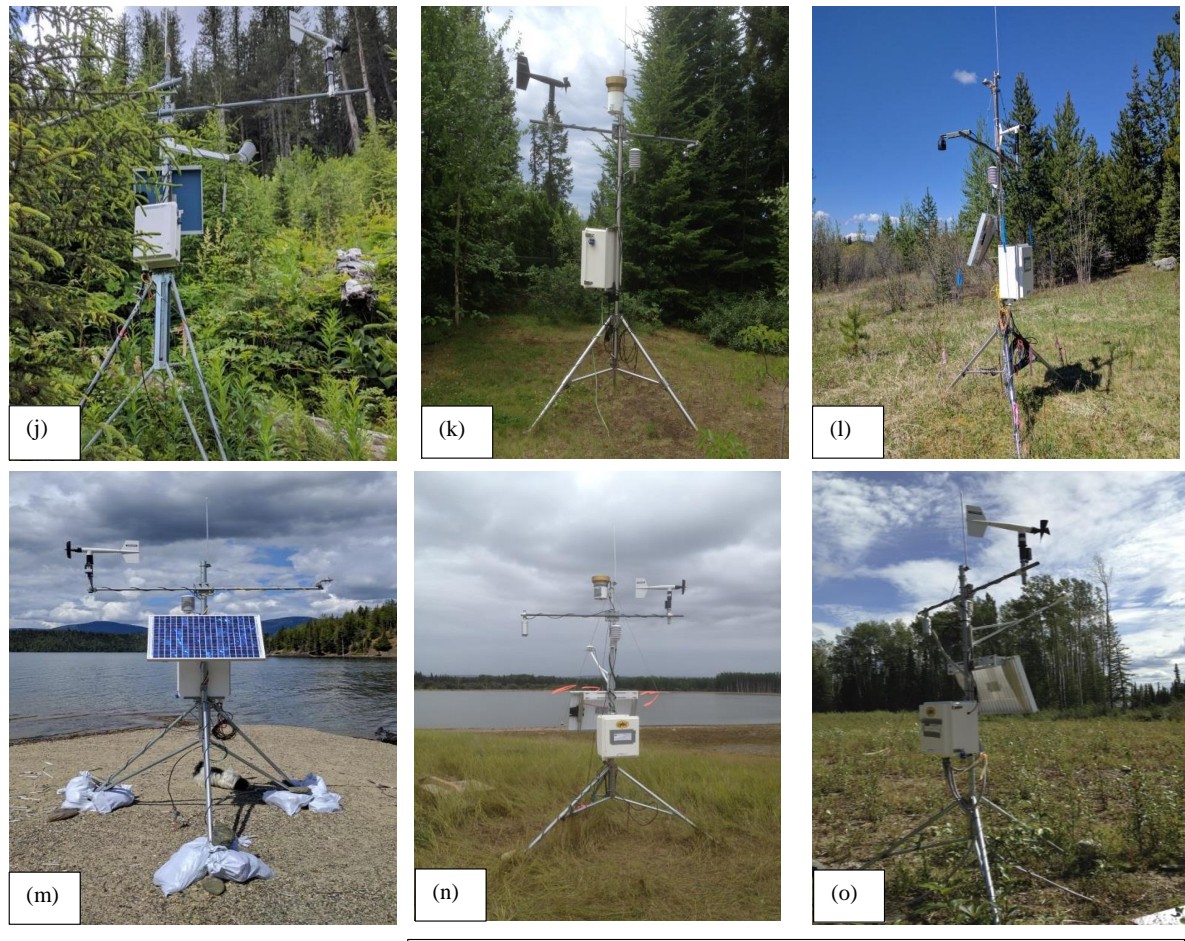

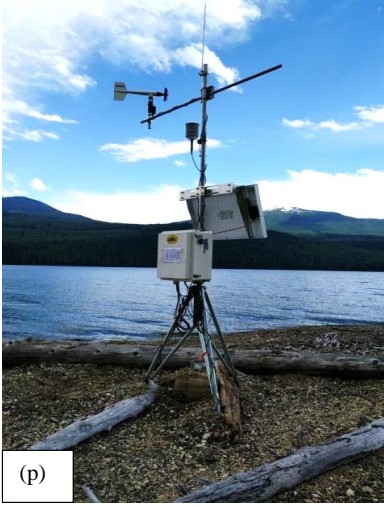

**Figure 4.** Photos of CAMnet weather stations: (a) QRRC in May 2017, (b) Spanish Mountain in June 2017, (c) Blackbear Mountain Radio Repeater in July 2015, (d) Browntop Mountain in June 2017, (e) Upper Castle Creek Glacier in September 2015, (f) Mount Tom in September 2008, (g) Lower Castle Creek Glacier in September 2015, (h) Castle Creek Glacier Radio Repeater in August 2009, (i) Ancient Forest in May 2017, (j) Lunate Creek in June 2017, (k) Ness Lake in June 2015, (l) Tatuk Lake in May 2017, (m) Plato Point in May 2017, (n) Kiskatinaw in August 2015, (o) Coles Lake in July 2015, and (p) Long Creek on June 2017.




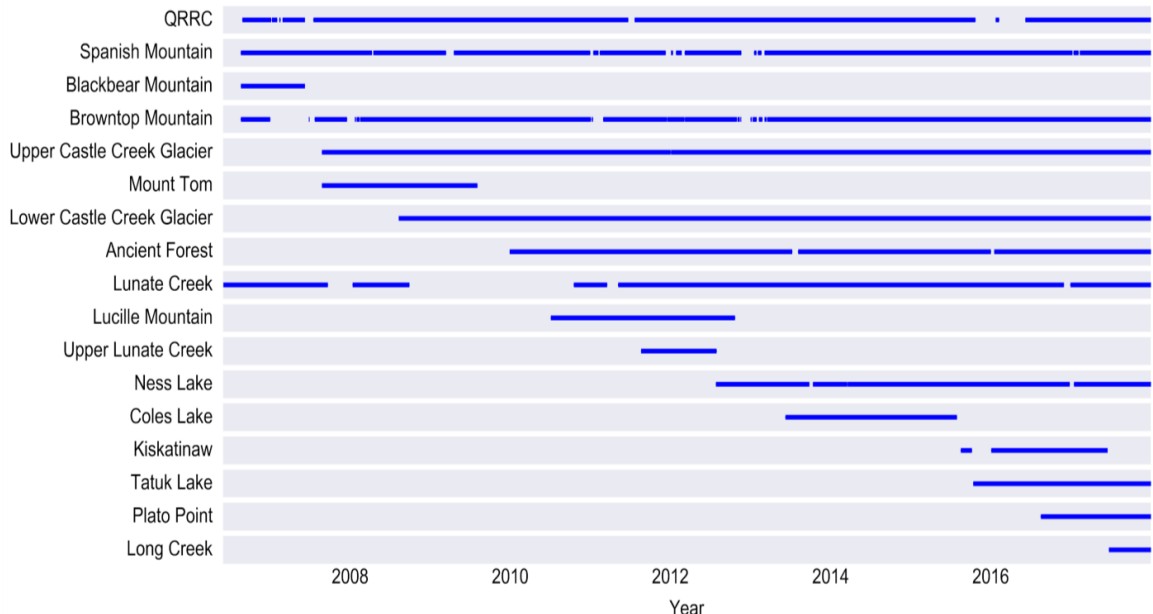

**Figure 5.** Operating periods for active and inactive CAMnet weather stations (1 January 2006 – 31 December 2017), deduced by the analyzing battery voltage required to power each station's datalogger and instruments. Gaps in datasets occur due to a variety of reasons outlined in Sect. 5.1. Note: Data may not be available for all instruments even when a station is operational (e.g., the icing of anemometers during winter).