# Peer review of "Manuscript under review for journal Earth Syst. Sci. Data"

_Earth System Science Data, 2018_

## Referee Comment (RC1) · Anonymous Referee #1 · 15 May 2018

I thought this was generally a clear and complete description of a nice mesonet data set. The manuscript, especially the abstract, needs some editing for grammar and punctuation. Inserting commas throughout the manuscript will make it more readable. While there are many mesonets in the world, the data from this one are an important contribution to the field. Data from automated stations in these types of environment typically have quality problems, so I hope the authors have performed a significant amount of quality checking. I'm not sure enough information has been provided for me, as a reviewer, to assess the data quality. I'd really like to see some time series

plots included in the data set, supplement, or manuscript. This would give me some indication of the data quality. The paper would also benefit from a climatology figure for temperature and rainfall, unless this has been published already (and could be reproduced here). The data is nicely organized. Does Zenodo commit to permanently archiving these data or are they permanently archived elsewhere?

---

## Author Comment (AC1) · 2 Jun 2018

RESPONSE TO ANONYMOUS REFEREE #1

We sincerely thank Anonymous Referee #1 for the constructive comments on our manuscript (Reference # ESSD-2018-45). We fully recognize and appreciate the reviewer's efforts in providing this informative report on our hydrometeorological dataset for the Cariboo Mountains of British Columbia (BC). Indeed, these insights will undoubtedly lead to an improved paper through this online discussion and ensuing revision process. We are thus taking into full consideration all of the comments from Anonymous Referee #1 and are preparing detailed responses to these as well as information on how the paper is being revised according to the referee's suggestions. A complete and detailed response document will be submitted once a decision has been made on our discussion paper. In the meantime, we provide here a general overview of our responses to the comments submitted by this referee in the following paragraphs.

Thank you for your general positive overview of our manuscript. We acknowledge the need to review the text carefully to improve grammar and punctuation (e.g., inserting commas where appropriate) throughout the manuscript including the abstract. We agree this mesonet plays an important role in filling a major observational gap in the otherwise poorly monitored Cariboo Mountains and surrounding areas of BC. Given the remote and often harsh environment in which the Cariboo Alpine Mesonet (CAMnet) weather stations are deployed, maintaining homogeneous and high-quality time series remains particularly challenging. Nonetheless, every effort is made in maintaining the integrity and homogeneity of the dataset and assessing its quality.

A concern raised by this referee is on the nature of the quality of the CAMnet data. First, we wish to point out that the equipment used is sourced from Campbell Scientific and its suppliers, and is considered the 'industry-standard' with high accuracy and extended operating ranges (please see Table 2 in the discussion paper). This equipment is also commonly used by various hydrometeorological networks across Canada including federal and provincial/territorial ministries (e.g., Environment and Climate Change Canada) and beyond.

Second, as outlined in Section 4.4 of the paper, data quality control and analysis is performed on all hydrometeorological data collected using a combination of automated procedures (e.g., codes in R) and visual observations. Recently downloaded data files are first checked to verify they have the total number of timestamps expected over the period of interest with missing ones in-filled with "NA". Otherwise, data values that are obviously in error are flagged and highlighted in yellow in the Excel spreadsheets archived on Zenodo. The variable with perhaps the most recurring data quality issues is snow depth as the data recorded by the sonic rangers are often spiky in nature due to interference of the acoustic waves with precipitation, birds, insects, etc.

CAMnet data compare favorably with measurements from independent meteorological networks such as those collected by Environment and Climate Change Canada, the BC Ministry of Environment, and the BC Ministry of Forests, Lands, Natural Resource Operations, and Rural Development (FLNRORD). As an example, monthly air temperature and precipitation collected at the BC Ministry of FLNRORD Likely Aerodrome weather station (52°36'54"N, 121°30'48"W, elevation 1046 m a.s.l.) correspond well with data from the CAMnet Quesnel River Research Centre (QRRC) weather station (52°37'06"N, 121°35'24"W, elevation 743 m a.s.l.)

with differences of only 0.45°C and 22% (117 mm) during 2014, respectively. Figures R1 and R2 show the temporal evolution of monthly air temperature and precipitation, respectively, recorded at the CAMnet QRRC weather station in Likely and at the BC Ministry of FLNRORD Likely Aerodrome weather station 5.2 km away. The evolution of both quantities matches very well. In response to this comment, we will add additional information on the quality control of the CAMnet hydrometeorological data and examples of comparisons with independent data such as those from the BC Ministry of FLNRORD.

[Figure]

**Figure R1:** Mean monthly air temperature at the CAMnet QRRC and at the BC Ministry of FLNRORD Likely Aerodrome weather stations in 2014.

[Figure]

**Figure R2:** Total monthly precipitation at the CAMnet QRRC and at the BC Ministry of FLNRORD Likely Aerodrome weather stations in 2014.

Additional comparisons of monthly mean minimum and maximum air temperature between the gridded ANUSPLIN dataset (McKenney et al., 2011) and the independent air temperature data at four CAMnet weather stations are reported in Sharma and Déry (2016). Their Table 2 shows high correlation values ($r \geq 0.97$, $p < 0.05$) and relatively low root mean square and mean absolute errors ($\leq 2.3$°C), with differences attributed in part to elevation disparities between the gridded and (in situ) point data. Of note, the QRRC, Browntop Mountain, Spanish Mountain and Upper Castle Creek stations all display coherent air temperature variability with ANUSPLIN data, suggesting the quality of CAMnet is well in line with other available products such as the widely used ANUSPLIN dataset.

Another point raised by the referee is the potential to add a climatology of air temperature and rainfall at the CAMnet weather stations. Despite the potential usefulness of the addition of climatological results to the paper, this would be beyond the scope of the present effort and the journal's purview. Instead, we refer the reader to other studies that have reported some statistics on the hydrometeorological variables collected at CAMnet stations (e.g., Déry et al., 2010; Sharma and Déry, 2016). A future effort will develop a comprehensive climatology of hydro-meteorological conditions recorded at all CAMnet weather stations and will be reported in a separate paper.

Finally, Zenodo does commit to permanently archive data stored on its platform but additional copies of the database are also stored on a computer server at UNBC. Zenodo's website states: "In the highly unlikely event that Zenodo will have to close operations, we guarantee that we will migrate all content to other suitable repositories, and since all uploads have DOIs, all citations and links to Zenodo resources (such as your data) will not be affected."

References:

Déry, S. J., Clifton, A., MacLeod, S., and Beedle, M. J., 2010: Blowing snow fluxes in the Cariboo Mountains of British Columbia, Canada, Arctic, Antarctic and Alpine Research, 42(2), 188-197.

McKenney, D. W., Hutchinson, M. F., Papadopol, P., Lawrence, K., Pedlar, J., Campbell, K., …, Owen, T. (2011). Customized spatial climate models for North America. Bulletin of the American Meteorological Society, 92(12), 1611–1622. doi:10.1175/2011BAMS3132.1

Sharma, A. R. and Déry, S. J., 2016: Elevational dependence of air temperature variability and trends in British Columbia's Cariboo Mountains, 1950-2010, Atmosphere-Ocean, 54(2), 153-170, doi: 10.1080/07055900.2016.1146571.

---

## Referee Comment (RC2) · Anonymous Referee #2 · 5 Jun 2018

Overall this paper is a good contribution to highlight the dataset they have compiled for an area of BC with limited observations, especially at higher elevations. Installing and maintaining stations in remote locations is very difficult and I applaud the authors in their efforts. In addition to maintaining the stations, compiling, QAQC'ing and making the data available is also a difficult task, and again the efforts of the authors to do this and provide these data to the scientific community are commendable. A general comment for the entire paper, is please show me some of the data! You cite examples of extreme windspeeds, wanting to capture differences based on elevation etc, yet you

do not provide example data. You also have some fairly long records at a few locations, show me some climate summaries too. As a potential user, it is very useful to see some summaries so I can decide whether the data are useful. For the most part, the paper has the pertinent information required, either in the main body or in the supplemental materials. I would recommend this paper for publication with minor revisions. Below are some more details by section, with technical recommendations at the end.

The INTRODUCTION is ok, but more information related to why this network is important is needed. What is the density of weather stations in the region? How does this compare to WMO standards, or other jurisdictions? Who can use these data (ie PRISM, PCIC, validating downscaled forecast data, Provincial and Federal agencies for fire and drought, flood forecasting, avalanche forecasting, hydrological modelling etc). The second paragraph is good as it shows how these stations have supported current and past research, well referenced.

The STUDY AREA section could use additional information. Please include information about seasonality, coldest and warmest months, maybe a precipitation and temperature graph by elevation? I need some context as a reader for the overall climate of the area (maybe even use some of your data!), or use ClimateWNA. Please list the biogeoclimatic zones and reference them. Since some of the focus is on the Quesnel River Watershed, you could also include a monthly hydrograph to provide context.

In the HYDROMET STATIONS Section, please provide some more information on the overview section such as how often sites are visited to download data and perform maintenance. Also explicitly say how many stations are in the Quesnel Watershed. This section should set up the Chrono Development section better so reader can easily identify where stations are located.

The CHRONO DEVELOPMENT section should be structured better to show how stations are geographically clustered ie have a subsection heading for Quesnel River Watershed, then describe stations, and then describe other stations as they fit together.

Consider renumbering the stations so they numerically match location ie all stations in Quesnel River Watershed are together ie 1 through 10, then the others from South to North, or some other logical grouping. When you list the stations in the paragraph, please include their elevation in brackets.

In the PRECISION AND ACCURACY section. I would like to see more information related to the calibration and how you checked for instrument drift – did you do field calibration for the tipping buckets and air temperature/RH? Were you proactive in swapping instruments ie change out every 2 to 3 years, or did you wait until they were broken. You note that sometimes old models break and cannot be replace exactly - I assume you try to replace with similar or betters specs? And also, how are you dealing with regeneration of forests and vegetation at your sites? Fairly common problem that many face, but hard to deal with. I would suggest trying to take upward fisheye photos or use drones to take an areal view at each site, good to track changes and also for users to understand the limitations of the data. ie is the trend due to climate change or encroaching veg?

Technical

Page 2, line 11-13 – please provide reference.

Page 3 line 1 – please give actual number of stations rather than "over a dozen".

Page 4 line 16, include link to QRCC web site. Page 6 line 22 – replace "sonic ranger" with SR50 Page 7 line 2 – name the station installed at Castle Creek Glacier. Page 7 line 7 – again name this station Page 7 line 8 – what heights were they measured Page 8 line 2 – reference Ancient Forest Page 13 line 23 – reference

In Figure 4, please include elevation of each station – I know this is elsewhere, but very handy piece of info to have. Figures should be stand alone, so this info is relevant.

---

## Author Comment (AC2) · 6 Jun 2018

RESPONSE TO ANONYMOUS REFEREE 2

We sincerely thank Anonymous Referee 2 for the constructive comments on our manuscript (Reference ESSD-2018-45). We fully recognize and appreciate the reviewer's efforts in providing this informative report on our hydrometeorological network and dataset for the Cariboo Mountains of British Columbia (BC). Indeed, these insights will undoubtedly lead to an improved paper through this online discussion and ensuing

revision process. We are thus taking into full consideration all of the comments from both Anonymous Referees 1 and 2. To that end we are preparing detailed responses to these as well as information on how the paper is being revised according to the referees' suggestions. A complete and detailed response document will be submitted once a decision has been reached on our discussion paper. In the meantime, we provide here a general overview of our responses to the comments submitted by this referee in the following paragraphs.

First, thank you for your general positive overview of our manuscript. We agree this mesonet plays an important role in filling a major observational gap in the otherwise poorly monitored Cariboo Mountains and surrounding areas of BC. Given the remote and often harsh environment in which the Cariboo Alpine Mesonet (CAMnet) weather stations are deployed, maintaining homogeneous and high-quality time series remains particularly challenging. Nonetheless, every effort is made in maintaining the integrity and homogeneity of the dataset and assessing its quality. Indeed, a considerable amount of effort has gone into the deployment and maintenance of the weather stations, in compiling and quality controlling the data, and in making the datasets available online; hence we thank the referee for acknowledging the extent of these efforts.

Second, in response to the referee's request, we will include a new subsection (4.6) under "Data collection" to provide some examples of the CAMnet data. Here we will demonstrate illustrative case studies of, for example, high wind events and/or strong inversions recorded at CAMnet stations. Another point raised by Referee 1, echoing comments from Referee 2, is the potential to add climate summaries for the CAMnet weather stations. Despite the potential usefulness of the addition of climatological results to the paper, this would be beyond the scope of the present effort and the journal's purview. Instead, we refer the reader to other studies that have reported some statistics on the hydrometeorological variables collected at CAMnet stations (e.g., Déry et al., 2010; Sharma and Déry, 2016). A future effort will develop a comprehensive climatology of hydrometeorological conditions recorded at all CAMnet weather stations

and will be reported in a separate paper. Nonetheless, as requested by the referee, we will provide a profile of air temperatures along a transect across the Quesnel watershed based in part on CAMnet weather station data (see later comment to this effect).

Thank you as well for recommending publication of the paper with minor revisions. The following paragraphs provide additional responses to comments on the various sections of the manuscript and the technical recommendations as well.

Introduction:

In revising the manuscript, we will provide additional information on the importance of CAMnet such as stating explicitly the density of active weather stations in this region (although that information can be derived from Figure 1 and the total area of the Cariboo Mountains (44,150 km$^2$)). Indeed, there are currently 49 active weather stations in the Cariboo Mountains, equivalent to a density of 1.1 station per 1000 km$^2$, well below the range of one station per 100-250 km$^2$ recommended by the World Meteorological Organization (Miles et al., 2003). This is also less than the observational density seen in other regions of Canada or globally (e.g., in Switzerland; Gubler et al., 2017). There are indeed many potential applications for the CAMnet data, including those mentioned by the referee, and we will add this information to the Introduction.

Study Area:

There is strong seasonality in meteorological conditions in the Cariboo Mountains and a graph illustrating a transect of climatological conditions will be added to the paper in the new Section 4.6. Information on the six biogeoclimatic zones found in the Quesnel River watershed will be added to the paper. These are: 1) Boreal Altai Fescue Alpine; 2) Interior Mountain – Heather Alpine; 3) Engelmann Spruce – Subalpine Fir; 4) Interior Cedar – Hemlock; 5) Sub-Boreal Spruce; and 6) Sub-Boreal Pine – Spruce (https://www.unbc.ca/quesnel-river-research-centre/quesnel-river-watershed).

While of interest, a hydrograph of the Quesnel River will not be added to the description
of the study area, as this information is readily available through other sources (e.g., Burford et al., 2009) and detracts attention from the primary focus and purpose of the paper.

Hydrometeorological Stations:

Additional information on the frequency of site visits will be added to this section. All CAMnet weather stations are visited at least once a year, but repeat visits are not feasible at the most remote sites (i.e. Upper and Lower Castle Creek Glacier weather stations) given access is often by helicopter. All other sites, however, are usually visited two to four times a year, while others such as the Quesnel River Research Centre (QRRC) and Ness Lake are easily accessible and thus serviced much more frequently.

As of 1 January 2018, there were five active weather stations (QRRC, Spanish Mountain, Browntop Mountain, Plato Point and Long Creek) in the Quesnel watershed, while two others (Upper and Lower Castle Creek Glacier stations) are just outside the watershed boundary. Section 3.1 will be revised to better set up the discussion that follows on the chronological development of CAMnet.

Chronological Development:

We respectfully disagree with the referee's comment in regards to the structure of this section, and will retain the existing one. Otherwise this would complicate the discussion given some stations were at times relocated from one site to another, such as the Blackbear Mountain weather station to the Upper Castle Creek site in 2007, just outside the boundary of the Quesnel watershed. Apart from the sites in and near the Quesnel watershed, there is no other geographical cluster of weather stations, and as such it would be difficult to rearrange this section as proposed. Keeping the numbering of the weather stations in a chronological order also facilitates tracking the development of the network.

Elevations for each of the stations is already provided in the text along with Table 1,

and such we refrain from duplicating that information in parentheses each time a specific location is sited in the text; however, we will add this information to Figure 4 as requested below.

Precision and Accuracy:

Calibration for most of the instruments including the tipping bucket rain gauges is performed by the supplier, Campbell Scientific and in our lab. We do attempt to be proactive in swapping instruments before they fail but often times this is not possible given tampering by animals or damage in extreme weather events. There is indeed the re-generation of vegetation at one field site, Spanish Mountain, which has led to an overall trend towards lower wind speeds there. That station therefore is only representative of a regenerating cutblock, which are common in the Cariboo Highlands. The suggestion of taking fish-eye photos or drones to take an aerial view of the sites will certainly be considered; note that we maintain an extensive library of digital photos taken at each site during each visit allowing comparisons of ambient environmental conditions over time including tree growth.

Technical:

Page 2, lines 11-13: The appropriate references will be added here (Hasler et al., 2015; Allchin and Déry, 2017; Beedle et al., 2015).

Page 3, line 1: The actual number of stations will be inserted here instead of "over a dozen".

Page 4, line 16: A link to the QRRC will be added (http://www.unbc.ca/qrrc).

Page 6, line 22: We will replace "sonic ranger" with "SR50".

Page 7, lines 2 and 7: The two stations near Castle Creek Glacier are named relative to their elevation, i.e. either 'lower Castle Creek Glacier' at an elevation of 1803 m a.s.l. or 'upper Castle Creek Glacier' at an elevation of 2105 m a.s.l. Given these are site names, we will henceforth use upper case letters when referencing them, i.e. 'Lower

Castle Creek Glacier' and 'Upper Castle Creek Glacier'.

p. 7, line 8: Temperature, humidity and wind speed are measured at 4.1 m and 2.7 m above ground at the Lower Castle Creek Glacier station.

p. 8, line 2: We will add Stevenson et al. (2011) as a relevant reference to the Ancient Forest.

p. 13, line 23: An appropriate reference for the radio signal range will be added here.

pp. 33/34, Figure 4: We will add the station elevations on this figure.

References:

Allchin, M. and Déry, S. J., 2017: A spatio-temporal analysis of trends in Northern Hemisphere snow-dominated area and duration, 1971-2014, Annals of Glaciology, 58(75pt1), 21-35, https://doi.org/10.1017/aog.2017.47.

Beedle, M. J., Menounos, B., and Wheate, R., 2015: Glacier change in the Cariboo Mountains, British Columbia, Canada (1952-2005), The Cryosphere, 9(1), 65-80, doi: 10.5194/tc-9-65-2015.

Burford, J. E., Déry, S. J., and Holmes, R. D., 2009: Some aspects of the hydroclimatology of the Quesnel River Basin, British Columbia, Canada, Hydrological Processes, 23(10), 1529-1536, doi: 10.1002/hyp.7253.

Déry, S. J., Clifton, A., MacLeod, S., and Beedle, M. J., 2010: Blowing snow fluxes in the Cariboo Mountains of British Columbia, Canada, Arctic, Antarctic and Alpine Research, 42(2), 188-197.

Hasler, A., Geertsema, M., Foord, V., Gruber, S., and Noetzli, J., 2015: The influence of surface characteristics, topography and continentality on mountain permafrost in British Columbia, The Cryosphere, 9(3), 1025-1038, doi: 10.5194/tc-9-1025-2015.

M. Miles Associates Ltd, for Ministry of Water Land and Air Protection, 2003: British

Columbia's Climate-Related Observation Networks: An Adequacy Review, Retrieved 5 June 2018 from http://www.for.gov.bc.ca/hfd/library/documents/bib94904a.pdf.

Sharma, A. R. and Déry, S. J., 2016: Elevational dependence of air temperature variability and trends in British Columbia's Cariboo Mountains, 1950-2010, Atmosphere-Ocean, 54(2), 153-170, doi: 10.1080/07055900.2016.1146571.

Stevenson, S.K., Armleder, H.M., Arsenault, A., Coxson, D., Delong, S.C., and Jull, M., 2011: British Columbia's Inland Rainforest; UBC Press: Vancouver, BC, Canada, 432 pp.